# LIPS - Learning Industrial Physical Simulation benchmark suite

**M. Leyli-Abadi, D. Danan, M. Yagoubi, S. Attoui**
IRT SystemX, Palaiseau, France

**A. Marot, J. Picault, B. Donnot**
RTE France, Paris, France

**P. Dimitrov, C. Etienam, A. Farjallah**
Nvidia

## Abstract

Physical simulations are at the core of many critical industrial systems. However, today's physical simulators have some limitations such as computation time, dealing with missing or uncertain data, or even non-convergence for some feasible cases. Recently, the use of data-driven approaches to learn complex physical simulations has been considered as a promising approach to address those issues. However, this comes often at the cost of some accuracy which may hinder the industrial use. To drive this new research topic towards a better real-world applicability, we propose a new benchmark suite "Learning Industrial Physical Simulations"(LIPS) to meet the need of developing efficient, industrial application-oriented, augmented simulators. To define how to assess such benchmark performance, we propose a set of four generic categories of criteria. The proposed benchmark suite is a modular and configurable framework that can deal with different physical problems. To demonstrate this ability, we propose in this paper to investigate two distinct use-cases with different physical simulations, namely: the power grid and the pneumatic. For each use case, several benchmarks are described and assessed with existing models. None of the models perform well under all expected criteria, inviting the community to develop new industry-applicable solutions and possibly showcase their performance publicly upon online LIPS instance on Codabench.

## 1 Introduction

Physical simulations constitute today a key enabler for real-world complex industrial systems (power grid management, rail infrastructure, aeronautics, pneumatic, gas production plants, thermal comfort, etc.), and are used at several critical stages of the system life-cycle (system design, solutions exploration, system V&V, etc.) to enhance decision making. Typically, the main drawback of using numerical simulations is their high computational cost to reach satisfactory solutions. It can become prohibitive for complex systems requiring large number of simulations. To tackle this issue, several techniques have been explored in the literature to design simplified physical models [1, 2, 3, 4], dimension reduction, or considering simplified assumptions to linearize the problem. In recent years, there has been a growing interest in using machine learning techniques to solve physical problems [5] for which conventional modeling approaches are very expensive to compute. The main goal is to accelerate the computation time while maintaining an acceptable accuracy of simulation predictions under some specified tasks. Going even further to reach the best trade-off, Deep Neural Networks (DNN) have recently led to promising results in various domains (see e.g.,[6, 7, 8, 9, 10]), allowing an important speed-up of simulations by substituting some computational bricks with data-driven numerical models.

36th Conference on Neural Information Processing Systems (NeurIPS 2022) Track on Datasets and Benchmarks.

These approaches emulate often existing simulators by learning from them in a supervised fashion and are generally used to complement them. Other approaches also aim at developing new kinds of differential solvers [11, 12, 13] in an unsupervised manner, and could possibly directly fit into existing simulator core. They often fall in the class of Physics-informed machine learning [14], where the learning is performed through a residual loss function and then physical constraints are verified on the learned model to validate the obtained solution. They could lead to stronger convergence and generalization than emulators. Another work in [15] provides also a taxonomy of integrating prior knowledge into learning systems. As automated learning of complex physical simulations is still considered as a new field of research, there exists a lack of common benchmarking pipeline, starting from available datasets, across various applications and finally common evaluation criteria as reviewed in section 3. This may allow to rigorously compare these methods and drive further advances into real-world applications, in particular when considering industrial use-cases.

In this paper, we propose a new benchmark suite "Learning Industrial Physical Simulations (LIPS)" to facilitate the use and the assessment of augmented physical systems, when applied on real-world applications. Depending on the application scope, the set of required physical variables to be considered may be different. The trade-off between computation speed and accuracy, as well as the expected generalization capability, may be specific to each industrial domain and the considered application. The compliance to physical laws of the learnt simulations may also be very important to validate them and consequently increase the user trust toward theses augmented simulators.

To develop the LIPS benchmark suite over several physical domains, we use a bottom-up approach by investigating two use cases described in section 2 with distinct physics: power grid and pneumatic. These 2 industrial domains both contribute in tackling ongoing real-world challenges, such as Climate Change, by transforming our energy system through electricity decarbonization and gains in transportation energy efficiency, or improving the decision-making efficiency regarding industrial products. They also allow, thanks to their heterogeneity in terms of physics lying behind modeling, a better assessment of our proposed benchmark. Preliminary ML models to benchmark also exist in the respective literatures. Our contributions, described in greater details in section 4, hence lie in:

1. defining application-oriented benchmark tasks for industry use cases as opposed to general-purpose simulation tasks;
2. proposing four categories of evaluation criteria that generalize to several physical, industrial and application domains and challenges beyond usual ML-only evaluation metrics;
3. sharing an open-source benchmarking suite framework (LIPS) with associated datasets;
4. opening a publicly available Codabench [16] thread providing a shared result table for user's submission and a fully automated and comparable evaluation.

Baseline experiments to demonstrate the usefulness of these benchmarks are run with existing state-of-the-art methods in section 5 and further discussed, highlighting the relevance of our benchmark.

## 2 Use-cases

### 2.1 The power grid case

**Industrial context**   Power System Operators are in charge of managing the security of large critical power grids (thousands electrical lines and substations that can be reconfigured) in real time and coordinate the supply and demand for electricity while avoiding fluctuations in frequency or interruptions of supply. It is of the utmost importance for a grid to be robust to blackouts at any time, which means in particular avoiding powerline overflows that can lead to a cascading failure (Figure 1, left). Operators have to face unexpected events (losing a line for example due to weather constraints) or to anticipate events such as variation of production during the day or as equipment's maintenance. They do so by assessing the risks, leveraging grid flexibility through simulations and carefully choosing sets of remedial actions which act on the grid topology or on the production levels.

**Applications**    Near real-time operations of a power grid can be classified into three steps with different expected speed and accuracy simulation trade-offs (Table 1):

1. *Risk assessment*, i.e., identifying problematic contingencies over a large number of possible cases while assessing their severity (anticipating for instance lines overloads, maintenance operations...);
2. *Remedial action search*, i.e., exploring for solutions to find a set of remedial actions on the grid such as topology change, to solve a local problem and assess its overall impact;
3. *Decision making*, i.e., selection and validation of one of the best solutions before implementation.

**Physical Simulations**    The computation of the grid state involves a set of physical laws (see appendix C.1) such as Kirchhoff's law or Joule effect. More specifically, the physical resolution of the problem is derived from a set of powerflow equations [17] described at any node $k$ of the grid. The power injected at a node of the network $s_k$ is the sum of active ($p_k$) and reactive powers ($q_k$): $s_k = p_k + q_k$. From Kirchhoff energy conservation law, the relation between voltage angle and magnitude can be formulated for node $k$ and neighboring nodes $m$ as follows:

$$\begin{cases} 0 = - & p_k + \sum_{m=1}^{K} |v_k||v_m|(g_{k,m} \cdot \cos(\theta_k - \theta_m) + b_{k,m}\sin(\theta_k - \theta_m)) \quad \text{Active power;} \\ 0 = & q_k + \sum_{m=1}^{K} |v_k||v_m|(g_{k,s} \cdot \sin(\theta_k - \theta_m) - b_{k,m}\cos(\theta_k - \theta_m)) \quad \text{Reactive power,} \end{cases} \tag{1}$$

where phasors $\boldsymbol{\theta_k}$ are unknown for all node $k$; either voltage $|\boldsymbol{v_k}|$ or reactive power $q_k$ are known input at any given node $k$; active power $p_k$ is a known input and $g_{k,m}$, $b_{k,m}$ known line characteristics for all nodes. For each line $l$, active $\boldsymbol{p^\ell}$ and reactive $\boldsymbol{q^\ell}$ powerflows or the current $\boldsymbol{a^\ell}$ can further be derived with Ohm's law.

**Significance**    The problem 1 is *non linear* and *non convex*. To estimate these variables, a Newton-Raphson power flow solver such as LightSim2grid [18] can be used. Over the past years, the required amount of simulations has drastically increased due to emerging trends [19] – mainly driven by Energy Transition initiatives, increasing renewable energy share as well as stronger exchanges with neighboring countries over the whole European grid, which lead to a greater stochasticity. In this context, the complexity of physical solvers becomes an obstacle for upgraded decision support [20]. An acceleration by several order of magnitudes is now expected.

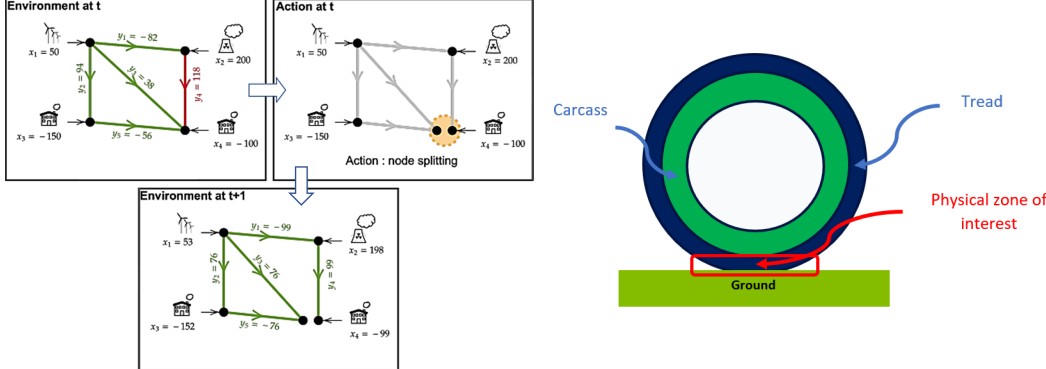

Figure 1: Left, illustration of line overloading in a power grid and a proposed solution (topology action) to overcome the cascading failure. Right, cross section of a simplified tire.

## 2.2   The pneumatic domain use case

**Industrial Context**    The tire performances have been improved over years with the aim to increase their resistance and to provide more comfort to driver's experience under various conditions. As such, their performance could be assessed with respect to the durability, ground adhesion or robustness. To do so, the tire behavior could be modelled during either rolling cycles or conditions such as crushing forces. Besides, to predict the global behavior, it is crucial to estimate the forces arising from the tire/ground interface which are concentrated on the contact area. Consequently,

to be able to optimize the underlying processes and deepen our understanding of the physical phenomena, we consider the conditions as near as possible to the real-world situations (considering the vehicle velocity, the tire pressure, the friction, the material behavior, etc.).

**Applications** Real-world tests performed on tires involve in particular two classical configurations depending on the accuracy/computational time simulation trade-offs (Table 1):

1. *Wheel sustaining*: assess whether the wheel is able to properly sustain the weight of a vehicle;
2. *Rolling cycle*: assess the behavior of the tire during the rolling phase.

**Physical Simulations** The computation of the tire state, involving the displacement and the contact stress arising from the contact conditions on the discretized domain (i.e. the mesh), is performed based on the resolution of a set of Partial Differential Equations (PDE) through Finite Element (FE) formalism (see appendix C.2 for more details). As such, the solution for the displacement is evaluated at all the nodes of the domain's mesh, whereas it is evaluated at the nodes on the contact boundary for the contact stress. In these PDEs, several physical considerations are involved: the behavior law, the relation between the stress acting on a body and the displacement, the motion law, the unilateral contact conditions (equivalent to assuming the ground is perfectly rigid), the Coulomb's law of dry friction. For more details about contact mechanics, the readers may refer to [21, 22, 23]. Note that, while the displacements and contact stress at the contact boundary are the actual unknowns of the problem, they may not represent all the required measures depending on the use case. Some physical quantities relevant to a given application can be computed as a post-processing of the unknowns. Figure 1, right, depicts the cross section of a simplified tire. We consider an idealized straight rolling on a non-deformable ground at constant speed.

**Significance** The problems mentioned above are strongly *nonlinear* due to the nonlinearity of the underlying behavior law, the large deformation framework and the frictional contact conditions. In order to estimate the displacement and the stress, the FE solver "Getfem" [24] is used. In practical applications, rolling simulations in particular provides a lot of useful information, such as the contact area, forces, contact pressure and moments. Classical methods exist [25, 26], however, because of the inherent complexity of the problem, the computation time is prohibitively expensive. Running over a day sometimes, it limits the use of such models in industrial applications compared to simpler surrogate model. An order of magnitude acceleration with acceptable accuracy would democratize its usage.

Table 1: Grid and pneumatic apps: speed vs accuracy and physical law compliance trade-offs

| | | Application | Variables to predict | Accuracy & PL compliance | Speed |
|---|---|---|---|---|---|
| **Use cases** | **Grid** | (1) Risk assessment | $a^\ell$ | + | +++ |
| | | (2) Action Search | $a^\ell$ , $p^\ell$ , $v_k$ | ++ | ++ |
| | | (3) Decision Making | $a^\ell, p^\ell, v_k, q^\ell, \theta_k$ | +++ | + |
| | **Tire** | (1) Wheel sustaining | $u_\Omega$ | +++ | ++ |
| | | (2) Rolling cycle | $u_\Omega, \lambda_c$ | ++ | +++ |

## 2.3 Added value of ML

Generally speaking, ML model can provide more direct and faster predictions than a Newton-Raphson resolution over the non-linearities of both use cases. It can leverage a learning memory of any given grid, mesh or last rolling cycle iteration of interest, without restarting the resolution from scratch as if it was a new system or problem. Additionally, in some of our benchmark tasks, we require the ML models to predict only a subset of the variables such as the flows or contact forces, unlike the physical solvers which usually compute all the variables by design. ML models could finally provide more factorized computation, such as, for instance, for varying grid topologies (varying number of electrical nodes), whereas existing physical solvers does not offer factorization over such dimension.

## 3 Related works and novelty

**Simulations and benchmarks in power grids.** Although, simulation time and convergence have improved over decades thanks to benchmarks based on shared power grid cases and some contests [27, 28], it remains still slow to compute large volume of simulations. In addition, existing simulators are general purpose and not application specific, that we consider in this work. Some application-oriented simulation-related benchmarks emerged lately in the power system community (SimBench [29], Power Grid Lib [30]). However, they are mostly designed to drive advances in operational research algorithms. In comparison, our benchmark: *a)* stresses the importance of considering the complexity of varying grid topologies for industrial applications; *b)* unlocks the creation of data-driven models by providing comprehensive data distributions to train them, similarly to [31] for other power grid related applications; *c)* defines specific metrics to evaluate them such as physics compliance, out-of-distribution generalization over unseen topologies or industrial readiness considering available data volume and scalability. It eventually allows a fair comparison of pre-existing ML models [8, 32, 33, 34, 35, 36] over all necessary dimensions as summarized in Table 2 and detailed in Appendix D. A similar evaluation over defined set of categories is also concurrently advocated by [37] as a first step towards proper benchmarks. Finally, we reference as an analogous initiative this recently published physical simulation-less but application-oriented dataset for power-grid ML [38].

Table 2: Comparative table between LIPS and related work for the power grid case.

| | Reference | Evaluation criteria categories | | | | Impact / Readability | Environment setup | | | |
|---|---|---|---|---|---|---|---|---|---|---|
| | | ML-related | Industrial Readiness | OOD Generalization | Physics Compliances | Thresholding & visualization | Dataset | Ref physical Simulator | ML model repository | Baselines |
| **ML model papers** | LeapNet [39] + [40] | Yes | Partial | Yes | No | No | Large, simple prod & varied topo distributions | Fast - Hades2 (proprietary) | Maintained | Yes (diverse) |
| | GNS/DSS [13, 35, 35] | Yes | No | No | Partial | No | Data generation, shared, simple distributions | Slow PandaPower (open source) | Not maintained | Yes (diverse) |
| | Fast Contingency analysis [33] | Yes | Partial | No | No | No | Not shared, realistic prod simple topo | Slow PandaPower (open source) | No access | Yes (diverse) |
| | Physics informed GNN [41] | Yes | No | Partial | Partial | No | Not shared, realistic prod simple topo | Slow PyPower (open source) | No access | Yes (uniform) |
| | Gridwarm [37] | Yes | Partial | No | Partial | No | Data generation, shared, simple distributions | None | Not maintained | Yes (diverse) |
| **Benchmark suites** | LIPS | Yes | Yes | Yes | Comprehensive | Yes | Large, doc, realistic prod topo distributions | Fast LightSim2Grid (open source) | Maintained | Yes (diverse) |
| | SimBench [29] | No (Optimization & heuristic) | Yes | Yes | No | No | Medium, doc, realistic prod simple topo | Slow PandaPower (open source) | No ML | Yes (diverse) |
| | PowerGridLib [30] | No (optimization) | Partial | Yes | Partial | No | Small, doc, realistic prod static topo | Med-speed PowerModel (open source) | No ML | Yes (uniform) |

**PDEs simulations and benchmarks for pneumatic.** In the last few years, the success of deep learning techniques has encouraged researchers to investigate their capability to solve PDE problems. Several works were proposed to hybridize PDE-based physical problems with Neural Networks (NN), from black-box resolution on unstructured meshes with graphs NN [42], to more interpretable approaches like the physics informed NN [14]. Some other works have focused on using un-supervised learning techniques to avoid the mesh construction (mesh-free methods)[43, 44]. Regarding pneumatic domain in particular, several attempts to use these techniques have already been made so far: the first tire/pavement contact-stress model based on artificial NN in [45] using a Neuro-Patch Model, tire modeling was investigated in [46] relying on a feedforward back propagation algorithm and [47] proposed a Structure-Preserving NN to predict the stress field within the tire. While providing promising results, none of these works attempt to compare fairly the performances of several ML models with respect to a set of significant application-based criteria and we propose to fill that gap. To our knowledge, this is the first ML-friendly benchmark for pneumatic.

**Benchmark for Learning to simulate physics.** Learning to simulate benchmarks started to emerge recently. The performance of a neural network architecture is studied extensively in [7] through several simulators based on different physical domains. Unfortunately, no resulting benchmark has been made available yet. It mainly relies on qualitative visual analysis, while more quantitative metrics as well as physical law verification could help for better comparison, as formalized later in this paper. The authors eventually claim that scalability to large systems remains currently an

issue, as well as the proper generalization in regions with high variability, highlighting the need for further advances. A new benchmark is also proposed lately in [48] over four PDEs canonical physical systems to drive forward the development of data-driven time integration solutions. Both focus primarily on scientific needs, with limited evaluation criteria categories, as opposed to industrial needs and applications.

**Identified research question (RQ).** In this paper, compared to above-mentioned related works, we address the following research questions: 1) There has been ongoing ML research for physical simulations for several years now. Are current evaluation setup comprehensive enough to actually provide applicable models in industry? If not, what is missing? 2) Can we define an homogeneous evaluation framework, with generic and comprehensive categories of criteria, for different industrial domains that could systematize the creation of such benchmarks and possibly drive cross-domain advances? 3) How can we represent an exhaustive set of benchmark results in an interpretable way?

We also set open research questions (ORQ) yet to be addressed that should be of interest for ML research: 1) What kind of inductive biases could help enforce ood generalization and physical consistency without sacrificing speed? 2) Is there a one-size-fits-all simulation model that performs best for all applications in a given domain or should it be more tailored to achieve better application-specific trade-off? 3) Could we foster the emergence of foundational models across domains?

Regarding RQ1, Table 2 shows the heterogeneity and weaknesses of current evaluation setups. Hence, a standardized and a comprehensive setup with meaningful categories and targets is needed, driving research towards industrial impact.

## 4 Benchmark suite design

### 4.1 Comprehensive evaluation criteria for benchmarking industrial physical simulations

The first step towards LIPS benchmark is a design of generic and yet comprehensive categories of evaluation criteria. It allows for a comparison within and across physical domains, while being expressive enough to represent industrial needs and expectations. ML-related only metrics are not sufficient in that regard. Thus, we introduce four categories of criteria of importance for industrial applications and illustrate their applicability and utility on 2 use cases in section 5.

**ML-related performance** Among classical ML metrics, we focus on the trade-offs of typical model accuracy metrics such as Mean Absolute Error (MAE) *vs* computation time (optimal ML inference time without batch size consideration as opposed to application time later).

**Industrial Readiness** When deploying a model in real-world applications, it should consider the real data availability and scale-up to large systems. We hence consider: 1) *Scalability:* the computational complexity of a surrogate method should scale well with respect to the problem size, e.g. number of nodes in power grid, mesh refinement level in pneumatic; 2) *Application Time:* as we are looking for a model tailored to a specific application, we measure the computation time when integrated in this application. To this end, we define a realistic application-dependent batch size, which may affect the speed-up.

**Application-based out-of-distribution (ood) Generalization** For industrial physical simulation, there is always some expectation to extrapolate over minimal variations of the problem geometry depending on the application. We hence consider ood geometry evaluation such as unseen power grid topology or unseen pneumatic mesh variations.

**Physics compliance** Physical laws compliance is decisive when simulation results are used to make consistent real-world decisions. Depending on the expected level of criticality of the benchmark, this criterion aims at determining the type and number of physical laws that should be satisfied.

### 4.2 Power grid application-oriented benchmarking task descriptions and datasets

From applications in Table 1, we define two application-oriented benchmarks. The Benchmark datasets depart from the same published realistic production and consumption distributions [49, 50], over two widely studied grids (IEEE 14 and IEEE 118 bus-systems) in the power system

literature [51]. However, each dataset has its own application-specific grid topologies (applied using Grid2Op [52] framework). The ground-truth for physical variables are further computed using LightSim2Grid [18], a physical solver with industrial-like performance on the selected grids.

1. **Benchmark 1 - Risk assessment through contingency screening.** The problem is to anticipate near real-time potential threats on the power grid and warn the operators accordingly [53]. It simulates incidents (aka contingencies) involving various elements of the grid (such as the disconnection of a line), one by one. For each contingency, a risk is identified when overloads on lines are detected. On a real grid, this scenario means running hundred of thousands of simulations, thereby, computation time is critical, especially since this risk assessment is refreshed every few minutes. We consider large simulation batches and the main physical variable is the line electric current $a^\ell$, because an overload occurs when it exceeds the line capacity.
   **Dataset specificity:** It presents grid snapshots including all possible line disconnections (N-1) for few different reference grid topologies. An ood topology test set containing N-2 line disconnections (2 line disconnections combined) is also attached to test for such generalization.

2. **Benchmark 2 - Remedial action search.** We need to explore possible solutions (aka "remedial actions") to identified risks for recommendation to the grid operator as in [54]. A solution consists in a predefined topological change on the grid that alleviates the previous overflow without generating any new problem. Those changes such as node splitting (see Figure 1) bring more non-linearity than line disconnections in benchmark1, making the distributions more complex. We here target medium-sized batches. Additional physical variables are predicted: active power flows $p^\ell$ and voltages $v_k$. A level of compliance with more related physical laws is expected. This allows the operator to better assess the system state in a difficult situation with some consistency.
   **Dataset specificity:** It presents grid snapshots when applying a topological reconfiguration (among a set of specified ones) on a single substation. It also considers some possible line contingencies that could cause overloads. An ood topology test set containing combination of 2 topological unitary actions is also attached to test for such generalization.

For more details about the datasets (input and output variables and their dimensions) for both industrial use cases, please refer to appendix C. Our "Datasheet for dataset" [55] in appendix A will also provide additional information concerning creation and contents of these datasets. For the power grid use case, we refer the readers to the Grid2op documentation [56]. A visual illustration of the baseline architecture is also provided in appendix F.2.

### 4.3   Pneumatic application-oriented benchmarking task descriptions and datasets

In this article, we focus on the tire mechanics concerning rigid surfaces. As shown in Table 1, we define two application-oriented benchmarks addressed in the literature, for instance in [47] for the rolling. To generate the datasets, we rely on the tire and experiment configurations described in [57]. Both the reference physical solution and the physical criteria of interest are computed by using the FE physical solver Getfem [24] and used as ground truth. Note that, the computation of pure mechanical criteria is performed by the physical solver for convenience, as their calculation rely on the underlying physical model at hand.

1. **Benchmark 1 - Wheel sustaining.** One of the basic function of a pneumatic tire is to support the vehicle weight. When a normal load is applied to a tire, it deflects as the load increases. Then, using the vertical load–deflection curves, we can estimate the so-called static vertical stiffness of tires. Such a criteria is known to have significant impacts on riding comfort, steering stability, and driving performance. Experimentally, this scenario implies running several simulations where different loads are applied on the wheel (inputs) to observe the resulting displacement of the structure (output). To be more specific, the physical variable we are interested in is the displacement $u_\Omega$.
   **Dataset specificity:** It presents displacement snapshots for different forces applied on the tire. Each displacement field arise from the simulation of a different static problem on a fixed axisymmetric mesh for the same physic.

2. **Benchmark 2 - Design testing during a rolling cycle.** We are also interested in assessing the behavior of the tire under the action of displacement-enforced rolling. Rather than the actual value of the evaluation criteria, we are interested about the relevancy of the design, i.e.

whether the values are within an acceptable range. Unlike the first scenario, this is a quasi-static configuration. Instead of running static simulations, a single quasi-static problem is run for several time instants within a time interval over rolling cycles. The physical variables we are interested in are the displacement $u_\Omega$ and the contact stress on the contact boundary $\lambda_c$.

**Dataset specificity:** It presents displacement and contact stresses snapshots evaluated at different instants during the rolling process. The idea is to train the model during $[0, t_1]$ and then evaluate the model for $t > t_1$; as such, it is a pure out of distribution example. Unlike the first case, it involves a single quasi-static problem on a fixed non-axisymmetric mesh with time as input variable.

## 4.4 Configurable benchmark suite architecture & ressources

Herein, we propose a unified extensible platform consisting of three modules combining data management, benchmark core and evaluation metrics. It allows the integration of all the previously mentioned benchmarks. The developed platform is flexible and allow to integrate more benchmarks from other similar domains. Note that this is different but complementary to NVIDIA Modulus framework [58]: one facilitates the design of PINNs models while ours focuses on benchmark design setup.

A Benchmark is instantiated by selecting a dataset, an augmented simulator and an evaluation object, as shown in Figure 2. Each module could be parameterized through a generic configuration file, which helps to relieve the burden for different setups and making it more user-friendly. They further comply with simple interfaces, making it modular to add new evaluation metrics or new physical domains.

**Benchmark resources** The benchmark implementation and corresponding data are provided in open-source via a github [1], alongside a starting kit aiming to facilitate the use of main functionalities. In

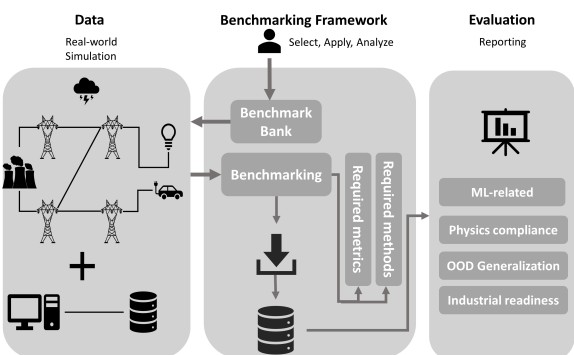

Figure 2: Benchmarking framework

addition, we make LIPS available on Codabench [59] – an open, public platform that allows to submit easily surrogate models, and to compare fairly submissions, under the same settings and in a fully automated way. The participant will also able to monitor their progress through a ranking table. We strongly encourage ML community working on physical problems to submit and evaluate their methods on previously-mentioned applications through the proposed platform. The public results could be highlighted and discussed at NeurIPS 2022.

## 5 Experiments

This section presents the evaluation results of baseline methods for each scenario of both use cases, alongside the experimental configurations used to obtain them.

### 5.1 Experimental setup

Regarding the stochastic nature of the optimisation methods based on gradient descent, 5 trials with different seeds has been executed and the performances reported based on mean and standard deviation of different runs. All the experiments in the following sections are performed using a server equipped with AMD EPYC 7502P 32-Core Processor, NVIDIA RTX A6000 GPU and 128 GB of RAM. All computation time evaluation are run on the CPU with time measured per simulation or prediction.

---

[1]`https://github.com/IRT-SystemX/LIPS`

**Power grid** - Our reference simulator LightSim2Grid has comparable speed to the proprietary RTE solver Hades 2 on mentioned IEEE grid cases, and is faster than PandaPower [60]. It is faster than the physical simulator used in SimBench, [29] by at least a factor 30 (see [18]) and also faster than the one used in Power Grid Lib [30] by at least a factor 5 on the hardware setup described above. Hence, the choice of this reference simulator makes our benchmark quite challenging. We have looked at a first baseline with differently tuned reference simulator. We set the maximum solver iteration to 1 to assess the maximum possible speed such solver can reach. Regardless of accuracy, we never go beyond a factor 5 speed-up. Hence, as it is far from expected speed-up, it has not been considered in further experimentation. But it definitely set a lower bound to outperform. We have then considered three different baselines for evaluation: a physics based simplification of power flow calculus which is DC approximation [61] and two augmented simulators which are Fully Connected (FC) architecture and a state-of-the art LEAP net [39], where contrary to FC, the topology intervenes in the latent space and demonstrate better combinatorial generalization capabilities. Note that we have conducted automated grid search to find the best performing network hyper-parameters for both architectures (see appendix F.4). Through this benchmark suite, we encourage the community to contribute and to suggest approaches aiming to improve the performances of the existing baselines.

**Pneumatic** - Our reference simulator Getfem is used to generate data in both benchmarks. Similarly to the Power grid case, we considered a simulation where only 1 nonlinear iteration is allowed for the underlying Newton algorithm used within the simulator. Putting aside the resulting loss of accuracy, the equivalent lower bound to outperform is close to 4.

We have considered two types of augmented simulators within the first benchmark: a FC architecture and a Unet [62] architecture. For the latter, the numerical solution evaluated by the physical solver on an unstructured mesh is projected on a $128 \times 128$ grid then, after the evaluation by the augmented solver, it is projected back to the mesh. For the second benchmark, two FC architecture are used: one to predict the displacement and one to predict the contact stress on the contact boundary.

### 5.2 Benchmark results and experiments

Table 3 summarizes the benchmark results for both use cases and their specific applications. In order to enhance the readability, we have made use of three qualitative levels from "not acceptable" to "great", relying on application-relevant threshold values reported in appendix C (tables 3 and 6). The full quantitative table from which this table is derived is also provided in section G.1 of appendix.

**Power grid**     As it can be seen it this Table, the ML based models (FC and LeapNet) show better accuracy for target variables than the baseline DC approximation. However, their performance on out-of-distribution dataset is still challenging and not acceptable. While the LeapNet shows a little better generalization performance, the accuracy is still above 6% error, on par with the reported performance in [39]. Maybe surprisingly, quantitative ood results on the small grid (inner small circles) are worse than the larger one. It can be explained by the fact that, in the smaller grids, any change has overall impact on all lines, hence it is even more challenging. The only possible physical law is also verified for this benchmark. Looking at a more complex benchmark 2, we can observe that further variables should be predicted and other laws should also be verified. The DC approximation respects most of the laws as it is based on physical solver, however it comes with some costs from the accuracy point of view. One order of magnitude speed-up when using ML models can be observed in comparison to a very optimized solver. Two order of magnitude speed-up would be expected at least on even larger grids. We emphasize that such speed-up time depends on the application context which needs to be considered. For more detailed comparison concerning the physics-based criteria, the readers may refer to appendix G.2.

**Pneumatic**     Likewise, regarding the pneumatic use case, it seems ML models perform relatively well in the first benchmark using FC architecture for the prediction of the displacement field, despite questionable results regarding the physics. The considered small dataset could also explain not acceptable obtained results using UNet architecture. Further investigations are required to assess its adaptability for this benchmark. For the second benchmark, despite the fact that it is a pure out-of-distribution case, the ML model behaves surprisingly well: the prediction is quite

Table 3: Benchmark result table for the two use cases under 4 categories of evaluation criteria. The performances are reported using three colors computed on the basis of two thresholds. Colors and symbol meaning: 🔴 Not acceptable  🟡 Acceptable  🟢 Great  ◎ two problem scales reported (in that case, speed-up for smaller scale is in parenthesis). The number of circles corresponds to the number of variables or laws that are evaluated. For quantitative values from which this table is derived, please refer to section G.1 of appendix and for a color blind version, please refer to Table 11 of appendix.

| | | | | Criteria category | | | | |
|---|---|---|---|---|---|---|---|---|
| | | | **ML-related** | | **Readiness** | **OOD Gen.** | **Physics** | |
| | | **Methods** | Quality | Speed-up | Speed-up | Quality | Domain laws | |
| Use cases — Power Grid — Bench1 | | DC | $a$ 🟢◎ | NA | 19 (7) | $a$ 🔴◎ | P1 🟢 | |
| | | FC | 🟢 | 19 (22) | 17 (20) | 🟡 | P1 🟢🟡 | |
| | | LeapNet | 🟢 | 17 (19) | 14 (17) | 🟡 | 🟢 | |
| Power Grid — Bench2 | | DC | $a$ 🔴  $p$ 🔴  $v$ 🟡◎ | NA | 5 (3) | $a$ 🔴  $p$ 🔴  $v$ 🟡◎ | P1🟢 P2🟢 P3🟢 P4🟢 P5🟢 P6🟢 P7🟢 P8🟢 | |
| | | FC | 🟢🟢🟡◎ | 99 (157) | 57 (27) | 🔴🔴🟡◎ | 🟡🟢🟢🟢🟢🟢🟢🔴 | |
| | | LeapNet | 🟢🟢🟢◎ | 90 (140) | 54 (24) | 🔴🔴🟡◎ | 🟢🟢🟢🟢🟢🟢◎🟢 | |
| Pneumatic — Bench1 | | FC | $u_\Omega$ 🟢 | 18 | NA | NA | P1🟡 P2🟡 P3🔴 | |
| | | UNet | 🔴 | 18 | NA | NA | 🔴🔴🔴 | |
| Pneumatic — Bench2 | | FC | $u_\Omega$🔴  $\lambda_c$🟢 | 11 | NA | $u_\Omega$🔴  $\lambda_c$🟢 | P1🔴 P2🟢 P3🟢 | |

accurate for the contact stress. The choice of an adapted scaler is important and could also influence the quality of displacement results. In all the investigated cases, the speed-up observed is at least one order of magnitude for both benchmarks compared to the physical solver, which was precisely our aim for the rolling case in the first place. However, given the accuracy for the displacement field, it is far from satisfactory and only partially met with our requirements.

We have shown with two very distinct industrial and physical domains that we can systematize the creation of comprehensive and yet homogeneous benchmarks for the use of physical simulation in industry, hence answering our RQ2. Our result table displays also a lot of benchmark outcomes, yet in a compact and readable way through the use of meaningful thresholds, colors and symbols: this answers our RQ3 and is an original benchmark result representation in the ML community to the best of our knowledge.

## 6   Conclusion and perspectives

This paper has investigated the definition and the implementation of a new benchmark suite, called LIPS (Learning Industrial Physical Simulations). We have addressed simulation-based industrial use-cases augmented with machine learning techniques. Two distinct industrial use cases (with different physics) have been considered to illustrate the proposed framework, with several application-oriented benchmarks. Experiments have shown several comparative studies based on proposed categories of criteria. The obtained results have also clarified the remaining challenges for existing state-of-the-art augmented simulators to emulate the behavior of a physical simulator in an industrial context. Although, they are much faster for providing the appropriate results, their interesting but yet insufficient out-of-generalization properties and vulnerability vis-à-vis the physics compliance highlights the requirement for further improvements. This benchmark opens the door for designing more robust and reliable augmented simulators that will find better real-world applicability. Future works will focus on extending the suite to new industrial use cases related to other physical domains (e.g. aeronautics, transport,...), which would help to improve the generalization of LIPS.

## Acknowledgments and Disclosure of Funding

This research was supported by IRT SystemX, RTE France, Michelin in the context of HSA project (hybridization of simulation and ML). Thanks to Reynaldo Gomez and Benoit Bastien, it was also supported by NVIDIA through an open collaboration to develop benchmarking framework and baselines for industrial physical simulation. We thank all contributors and colleagues from RTE, Michelin, IRT SystemX and NVIDIA who provided insight and expertise that greatly assisted the research. For the pneumatic usecase, we would like to thank Raphaël Meunier, from Michelin, for his involvement regarding the usecase design and relevant industrial applications advices. We are finally very thankful to EDF Exaion for sponsoring through their infrastructure these developments by making GPU servers available for training all baselines as well as for running the Codabench public evaluation thread.

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
