# OpenReview forum: "LIPS - Learning Industrial Physical Simulation benchmark suite"
_NeurIPS.cc/2022/Track/Datasets_and_Benchmarks — NeurIPS 2022 Datasets and Benchmarks _

### Official Review · Reviewer_vNMc · 2022-07-25
**Paper Review**

**Rating:** 6
**Confidence:** 2
**Clarity:** Yes.

**Strengths:**

This paper focuses on specific industrial tasks rather than general physical simulations, which benefits the development of ML algorithms in industrial applications.

**Weaknesses:**

No clear comparison between the proposed simulator and previous ones. According to Section 3, there are some similar simulators such as SimBench and Power Grid Lib. A table that lists all features and advantages of these simulators would make the comparison clearer.



**Additional Feedback:**

None.

**Correctness:**

This paper proposes 4 sets of metrics, including ML-related performance, Industrial Readiness, Application-based out-of-distribution Generalization, and Physics compliance. According to the description of these metrics, the proposed benchmark is appropriately designed.

**Documentation:**

The documentation is sufficient and clear.

**Ethics:**

No.

**Relation To Prior Work:**

A table that lists all features and advantages of these simulators would make the comparison clearer.

**Summary And Contributions:**

This paper proposes a simulation for industry use cases (e.g., Power System Operators) as opposed to general-purpose simulation tasks and 4 categories of evaluation metrics. They also provide an open-source benchmarking suite and a public leaderboard for user submission and a fully automated and comparable evaluation.

---

> ### Author Response · Authors · 2022-08-08
> **Anwsers to reviewer**
>
> We appreciate your time and effort for reviewing our paper and capturing the overall aim and specificity of our benchmark suite.
>
> Regarding the comparison to previous physical simulators, we have already made the comparison with our reference physical simulator of choice. We agree it is a good sanity check to also have in mind other physical simulators around, but it will not make the benchmark more challenging. As this is a several-decade old problem, all such powerflow simulators have reached expected accuracy on those reference IEEE power grids (which might not be the case for PDE-like problems) and the only remaining discriminative feature is computational speed.
>
> In our case, we have actually chosen the best open-source performing simulator with industrial-like performance (which we know of by comparing its computational speed with internal Hades2 RTE proprietary solver with similar performance on those test cases), but we will better explain it in the paper.
>
> Note that those two other benchmarks (Sim2Bench and Power Grid Lib) deal with somewhat different tasks, not specifically geared towards physical simulations and more concerned with quality or consistency of solutions rather than speed. We have referenced them as interesting benchmarking initiatives in the domain of power system but with a different scope. However they indeed have connections to an underlying powerflow simulator.
> - For Sim2Bench, which targets  grid analysis planning problems rather than near-real-time operations as we do, the related powerflow simulator is pandapower. Simulation for them relates to grid planning scenario simulation, less specifically grid state physical simulation. Our reference LightSim2grid already have already been benchmarked towards pandapower as it inherits from it but is way faster (more than 30 times speedup) as compiled in C++ (see https://github.com/BDonnot/lightsim2grid#benchmarks).
> - For Power Grid Lib, this is a benchmark based on PowerModel library in Julia dedicated to overall optimization problems, in which they cast the physical equations but don't reuse an existing physical simulator per say. They also provide a powerflow simulator; although we do not have a precise estimate of its computation time yet (we may provide it if required), we estimate that the speed up factor of lightsim2grid is about 10.In brief, for the power grid case, we took the fastest physical solver (according to our knowledge) as a reference, in order to have a solid baseline for comparison with ML surrogate models. If in the future new physical solvers perform better, it will be also possible to assess their performance easily through our benchmarking platform.
>
> Would this kind of discussion be enough to include in our paper to ensure that we have selected the best reference physical simulator so far in our benchmark ? Would you still like to see their speed compared in a Table ? Or added in our Benchmark result table ? Basically physical simulators will have all green dots in the table and will only differ in the speed-up factor.
> Note that following a suggestion from Reviewer 2, we will add a baseline result of the physical simulator with a different tuning where we loosen its accuracy to see what speed up we get.

---

> > ### Comment · Reviewer_vNMc · 2022-08-24
> > **Thanks for answering my question**
> >
> > I appreciate the authors provide comparison between their simulator and two other simulators. It would be great to add this discussion to the manuscript. I don't have further questions but I will keep my score since I am not familiar with industrial simulation and application.

---

> > > ### Author Response · Authors · 2022-08-25
> > > **Comprehensive comparative table with related work to highlight our unique contribution**
> > >
> > > Thanks for your feedback, and we are glad that our response answered your concern. We have tried to make our paper as accessible as possible to people not expert in the industrial domains, as we think NeurIPS reviewers should not have to be, yet it could still be improved. And actually thanks to your suggestion of making a comparative table, we decided to push our response even further to really highlight our unique and unifying contribution in comparison to the related work we cited for non-domain experts. We wanted to give you some credit for pushing us that way, we think it better demonstrates what our work stands for and we hope you will appreciate it. A colorful table for improved readibility is included in our revised paper
> > >
> > > We are also integrating explicit research questions not specific to the industrial domains that you might find of interest for improved clarity.
> > >
> > >
> > >
> > > |                  |                                | evaluation criteria categories |                      |                     |               | impact                  | readability              | environnement setup                                |                                      |                     |                 |
> > > |------------------|--------------------------------|--------------------------------|----------------------|---------------------|---------------|-------------------------|--------------------------|----------------------------------------------------|--------------------------------------|---------------------|-----------------|
> > > |                  | **paper reference**                | **ML-related**                   | **industrial readiness** | **ood generalization**  | **PL compliance** | **meaningful thresholding** | **standard comparative viz** | **dataset**                                            | **ref Physical simulator**              | **ML model repository** | **baselines**      |
> > > | **ML-model papers**  | LeapNet [51]                   | yes                          | no                   | yes                 | no            | no                      | no                       | large, simple prod distributions                   | fast - Hades2 (proprietary)          | maintened           | yes & diverse   |
> > > |                  | GNS/DSS [13,31,34]             | yes                          | no                   | no                  | partial       | no                      | no                       | data generation shared, simple distributions       | slow - PandaPower (open source)      | unmaintained        | no              |
> > > |                  | Fast contingency analysis [32] | yes                          | partial              | no                  | no            | no                      | no                       | not shared, realistic prod but simple topo         | slow - PandaPower (open source)      | no access           | yes & diverse   |
> > > |                  | Physics-informed GNN [35]      | yes                          | no                   | no                  | no            | no                      | no                       | not shared, realistic prod but simple topo         | none                                 | no access           | yes but uniform |
> > > |                  | Gridwarm [36]                  | yes                          | partial              | no                  | partial       | no                      | no                       | data generation shared, simple distributions       | none                                 | unmaintained        | yes & diverse   |
> > > | **Benchmark suites** | **LIPS**                           | yes                          | yes                  | yes                 | comprehensive | yes                     | yes                      | large & doc & realistic prod + topo distributions  | fast - LightSimgrid (open source)    | maintained          | yes & diverse   |
> > > |                  | SimBench [28]                  | no(otimization & heuristics) | yes                  | yes                 | no            | no                      | yes                      | medium & doc & realistic prod but simple topo      | slow - PandaPower (open source)      | no ML               | yes & diverse   |
> > > |                  | PowerGridLib [29]              | no (optimization)            | partial              | yes                 | partial       | no                      | no                       | small & doc & realistic prod but static topo       | med-speed - PowerModel (open source) | no ML               | yes but uniform |

---

### Official Review · Reviewer_vgwR · 2022-07-25
**Reviews for industrial simulation learning paper**

**Rating:** 4
**Confidence:** 3

**Strengths:**

**Significance**:
The paper considers an interesting industrial simulation learning problem, and presents two valuable datasets for learning the simulation for power grid and the pneumatic types. The datasets might be useful in a broad AI4science community and could facilitate the relevant research.


**Accessibility**:
The paper provides the link to the code and also offers a nice way to compare different methods via Codabench.

**Weaknesses:**

**Significance**: [the challenges posed by this benchmark is unclear]

Although the paper offers a good discussion of the need for fast physical simulation and presents the two datasets for learning such physical simulation, it is still unclear what the challenges this benchmark is targeted at. Specifically, the datasets themselves have not been described to reveal the underlying research questions. For example, for Risk assessment through contingency screening benchmark, it seems that the major challenge is the out-of-distribution (OOD) generalisation. But it is not quite clear what “OOD topology test set” (line 227) really is and how it differs from the training set (in terms of OOD). It would be nice to provide more analysis about the dataset difference and the OOD issue. For other benchmark tasks, e.g., Remedial action search and  Static vertical stiffness, it is not even clear what the tasks are from the machine learning perspective, let alone the challenges posed by the dataset.


**Relevance**: [the relevance to the machine learning research is unclear]

The baseline methods proposed in the paper do not sufficiently showcase the difficulties of the benchmark tasks. It instead poses another question as to how relevant the benchmark is to the machine learning research. For example, a fully-connected (FC) architecture is used for the power-grid benchmark, whereas a convolution architecture is used for pneumatic. It looks like different neural network architectures (an engineering choices, not a research question) may play a crucial role. Would it be possible to try more heuristic methods (e.g., different architectures) on both datasets? Moreover, it is unclear how the proposed evaluation metrics can be quantified, e.g., “Scalability: the computational complexity of a surrogate method should scale well depending on the problem size” (line 198 - 199), "extrapolate over minimal variations of the problem geometry” (line 204), and “the expected level of criticality of the benchmark” (line 208)


**Accountability**: The current version of the paper raises some questions about the dataset details: from the machine learning perspective, what are the input and output for each dataset? How large is the dataset size? How the training and testing dataset is split? Why U-net is used as a baseline for pneumatic dataset?


The current version of this paper may not be qualified as an accept. If the paper could present the dataset and the benchmark with clearly defined research questions and more convincing baselines, it would be a good paper. Given the time constraint in the rebuttal period, it might be hard to address all above issues. So I tend to reject this paper for its current version.

**Additional Feedback:**

n/a

**Clarity:**

The paper is written well in general. But I have some questions:

* Table 2: Circles in the table title and in the table are in different colors.

* Table 2: it is unclear what it means by “two problem scales reported”

* Table 2: what is the meaning of these numbers? e.g., 19(7).

* Section 4.1 Comprehensive evaluation criteria: how should the Industrial Readiness, particularly “Scalability: the computational complexity of a surrogate method should scale well depending on the problem size”, be measured?

**Correctness:**

The claims are correct. However, it is unclear how the dataset is constructed (e.g., input and output, data structure etc.). the evaluation methods are a bit weak and the evaluation metrics are not quite clear.

**Documentation:**

yes

**Relation To Prior Work:**

yes

**Summary And Contributions:**

The paper introduces a set of datasets for learning the industrial physical simulation. The proposed datasets are targeted at two real-world applications: power grid and pneumatic simulation of tires. The power-grid dataset consists of two benchmark tasks: Risk assessment through contingency screening & Remedial action search. The pneumatic-simulation dataset also consists of two benchmark tasks: Static vertical stiffness simulation & Design testing during a rolling cycle. To evaluate the performance on these benchmark tasks, the paper also proposes metrics: ML-related performance, Industrial Readiness, Application-based out-of-distribution Generalization, and Physics compliance. The paper finally presents several heuristic baselines to show the unsatisfactory	performance of current methods, which calls for a family of new methods.

Overall, this paper presents a set of promising benchmark tasks for industrial simulation but lacks sufficient details and baseline methods to support the claims.

---

> ### Author Response · Authors · 2022-08-08
> **Answers to reviewer - request for initial discussion**
>
> Thank you for your time and very detailed review. We are glad that overall you consider the topic of interest for ML research, mentioning a possible connection to AI4science kind of initiative. Yet we understand that you are looking for more details and clarifications. We hence would like to open the discussion on your high-level concerns as a first step if this is fine with you. We will provide detailed answers to more specifics remarks after this initial discussion.
>
> Regarding the **datasets**, we would have indeed liked to put more details in the main paper. But because of limited space, we eventually put most of them in the supplementary material, completing a Datasheet for dataset (with information of dataset size for instance) and providing a "Benchmark Physical laws and variables" section where inputs and outputs are detailed. We also provided there one baseline architecture (LeapNet) schematic which shows inputs and outputs. For the power grid, the dataset was generated with the grid2op framework which documentation also describes the variables in the dataset https://grid2op.readthedocs.io/en/latest/observation.html#main-observation-attributes
>
> Could you confirm this is the kind of information you were looking for ?
>
> Thanks to your feedback, we will also add more details such as the different variable dimensions and maybe put some of this content back in the main paper (similarly to Table 1 in "An Extensible Benchmark Suite for Learning to Simulate Physical Systems [43]"), given that we have one additional page we can use now.
>
> Regarding **research questions**, we have tried to highlight the current gaps in "related works and novelty" section 2 and how we aim at answering them. Phrasing corresponding research questions more explicitly, our work is concerned with:
> - There has been ongoing ML research for physical simulations for several years now. Are current evaluation setup comprehensive enough to actually provide applicable models in the industry ? If not, what is missing ?
>     - This echoes “Challenges in deploying machine learning: a survey of case studies” paper and others
> -  Can we define an homogeneous evaluation framework, with generic and comprehensive categories of criteria, for different industrial domains that could systematize the creation of such benchmarks ? And possibly drive cross-domain advances, reusing similar architectures, or possibly foster the emergence of overall foundational models?
> - How can we represent an exhaustive set of benchmark results in an interpretable way ?
> - Open research question: What kind of inductive biases could help enforce ood generalization and physical consistency ?
> - Open research question: Is there a one-size fits all simulation model that performs best for all applications in a given domain or should it be more tailored to achieve better application-specific trade-off ?
>
> Does this help clarify the scope of our work ? Do you have any further suggestions on that matter ?
>
> We want to emphasize that we put much of our conceptual effort into our generalization work, that is the definition of the set of evaluation criteria and their effective representation in the benchmark result table. We would be keen to hear any more feedback or suggestions from you on this as well.
>
>
> In terms of **baselines**, for the power grid use case we have actually selected one reduced-ordered model + one standard Fully connected Neural Network (NN) + one state-of-the art NN (LeapNet, an architecture that aims at improving ood performance by design).
> Would you agree that such a diverse set of baselines is relevant enough to show the difficulty of our benchmark ? We will nonetheless include one **additional baseline** as suggested by reviewer 2.
>
> Note that for each NN type we also have conducted a neural grid search **to find the best performing architectures** (please refer to the"hyper-parameter tuning" section of supplementary material but we will add more details to it). This shows that solving the benchmark is not simply a matter of engineering but rather of model design breakthrough to not only be fast and accurate at interpolation, but also consistent and accurate with some expected extrapolation capability. Note that we have quite a deep knowledge of Deep Learning for power systems developments with RTE (French power system Operator) and this benchmark is the result of many gaps we have observed in related research works since then, missing critical points for industrial applicability.
> For the pneumatic use case (which is more novel in terms of existing DL developments), we are working to provide a similar set of baselines, which were a bit weaker indeed in that case.
>
> Based on your additional feedback, we will improve our work and further answer the remaining and more specific concerns you raised.

---

> > ### Author Response · Authors · 2022-08-22
> > **Request for feedback**
> >
> > Dear reviewer,
> >
> > before the review process ends this week, please could you kindly provide us some feedback on the discussion we initiated in our previous comment?
> >
> > This will be very helpful to complete the writing of the final version of the paper.
> >
> > As a summary, our main questions were:
> > - regarding datasets: *are you satisfied with the level of details we provide in the supplementary material?*
> > - regarding research questions: in our previous comment we identified a set of clear research questions such as: "Is there a one-size fits all simulation model that performs best for all applications in a given domain? How can we represent an exhaustive set of benchmark results in an interpretable way? What kind of inductive biases could help enforce ood generalization and physical consistency?" *Does this phrasing helps clarifying our work? Or do you have any further suggestions on that matter?*
> > - regarding baselines: *do you agree with the relevance of the choices we did about NN architectures as explained in ourprevious comment?*
> >
> > Thank you very much for your help!
> >
> > Best regards,
> >
> > The authors.

---

> > > ### Author Response · Authors · 2022-08-25
> > > **Answers to the different concerns raised (1/2)**
> > >
> > > Regarding the dataset descriptions
> > > ```
> > > The current version of the paper raises some questions about the dataset details: from the machine learning perspective, what are the input and output for each dataset? How large is the dataset size? How the training and testing dataset is split?
> > > ```
> > > We have better referenced to the relevant appendix sections with "datasheet for dataset" and tables describing inputs and outputs (to which we added variable dimensions). We have added a link also to the relevant Grid2op documentation that we use for the power grid use case. An example model architecture in the appendix also  shows inputs and outputs.
> > >
> > > Considering the benchmark task descriptions
> > > ```
> > > For other benchmark tasks, e.g., Remedial action search and Static vertical stiffness, it is not even clear what the tasks are from the machine learning perspective
> > > ```
> > > We have referenced the input and output variables, as well as the ood variables in the above mentioned table of the appendix. We have also added reference from the literature for further understanding of the tasks. For Remedial action search, we have also pointed to Figure 1 to illustrate what is meant by remedial action.
> > >
> > > Regarding the research questions, we have now included 3 questions that we respond to in this paper as well as 3 open-research questions to the ML community that our paper can contribute to answer.
> > >
> > > About the baselines, with the addition of the differently tuned physical simulator, we think we have covered a relevant and diverse set of baseline including one reduced ordered model, one standard neural network architecture, one state-of-the-art LeapNet architecture. None actually achieves expected speedup while enforcing satisfying ood generalization and physical consistency. If other models actually exists in the literature, none actually reported performance on all those necessary criteria in the literature, and no other model beside LeapNet is both shared and maintained to be reused. But with our deep knowledge of this litterature, we are confident that none actually achieves satisfactory performance on ood generalization, and while some might achieve physical consistency it comes at the cost of a unsatisfactory slowdown.
> > > ```
> > > it seems that the major challenge is the out-of-distribution (OOD) generalisation. But it is not quite clear what “OOD topology test set” (line 227) really is and how it differs from the training set (in terms of OOD). It would be nice to provide more analysis about the dataset difference and the OOD issue
> > > ```
> > > Indeed this is a major challenge, and it related to combinatorial generalization: how from the knowledge of single intervention in a power grid graph, you generalize to a combination of such interventions. This kind of OOD generalization was initially proposed in the LeapNet paper which already provide a good set of information. We have added in the input/output table of the appendix, a mention to the variable concerned with ood combinatorial generalization. We are in the process of integrating a visualization package (Grid2Bench) of this dataset with LIPS and will soon be able to provide additional distribution visualizations of training vs ood test set.
> > >
> > > Regarding the pneumatic use case, ML research have been scarce so far and no SOA model really exist. Beside fully-connected, we have favored Unet as it is a rather straightforward and established model today that can handle some invariance of the problem. It can be applied if you consider that the displacement field can be represented on a 2D images. In this rebuttal phase, we have tried PINN approaches, but we have been unsuccessful so far to provide better results. We see that as a good perspective for the ML community to provide such improved models. We have nonetheless considered the additional baseline of differently tuned physical simulator but which achieves too limited speed-up.
> > >
> > > ```
> > > For example, a fully-connected (FC) architecture is used for the power-grid benchmark, whereas a convolution architecture is used for pneumatic. It looks like different neural network architectures (an engineering choices, not a research question) may play a crucial role.
> > > ```
> > > Actually FC are used for both use cases as a standard baseline. For the power-grid, LeapNet is used as a state-of-the art model, and Unet is used for the pneumatic with the considerations mentioned above. We tried as much as possible to look for available SOA models from the literature, not really with engineering considerations. Also we have used automatic libraries to avoid unfair fine-tuning of network hyperparameters, not engineering this particular aspect. With current available SOA models, we think that we showed that our benchmarks are still an open challenge, and some might take some years to reach a great trade-off.

---

> > > > ### Author Response · Authors · 2022-08-25
> > > > **Answers to the different concerns raised (2/2)**
> > > >
> > > > About
> > > > ```
> > > > it is unclear how the proposed evaluation metrics can be quantified, e.g., “Scalability: the computational complexity of a surrogate method should scale well depending on the problem size” (line 198 - 199), "extrapolate over minimal variations of the problem geometry” (line 204), and “the expected level of criticality of the benchmark” (line 208)
> > > > ```
> > > > For extrapolation over minimal variations of the problem geometry, the metric are based on accuracy and then it is up to the domain expert to include in its ood test set the relevant variations he wants to consider and test over. For instance combinatorial generalization over graph topology changes for the power grid, or different resolution (space or time) for the pneumatic.
> > > > For the expected level of criticality of the benchmark, it is up to the domain expert to assess this, hence deciding in a given task to put more or less weight on physical compliance and accuracy versus speed-up. For example in the power grid operations, risk assessment (further away from real-time) is a less critical stage than validation of implementing an action (in real-time with direct consequences).
> > > >
> > > > Regarding the scalability, it is trickier to come up with a single measure. In our result table, we have decided to super-imposed results for diferrent power grid sizes, leaving it up to the table reader the interpretation of the scalability. Yet if we could have had a discussion on this, we would have maybe had a proposition. Also related to
> > > > ```
> > > > how should the Industrial Readiness, particularly “Scalability: the computational complexity of a surrogate method should scale well depending on the problem size”, be measured?
> > > > ```
> > > > For the scalability regarding speed-up, we could have had a indicator depending on the scaling power law: 0 (if rootquared law) - 30 (if linear law) - 60 (if quadratic law) -100 (if above cubic law)
> > > > For the other criteria scalability, we could have count the proportion of indicators that would not have worsened on the larger scale compared to other scale. Defining such single measure could be a perspective that would simplify even further our result table.
> > > > Yet as the result table illustrates, we were able to quantify all evaluation criteria for the considered use cases.
> > > >
> > > > Finally
> > > > ```
> > > > Table 2: Circles in the table title and in the table are in different colors.
> > > > ```
> > > > Thank you we will correct this
> > > >
> > > > ```
> > > > Table 2: it is unclear what it means by “two problem scales reported”
> > > > ```
> > > > For the power grid, two different grids of different size were studied. We hence report the results for those two grids: inner circle is for the small grid and larger circle for the large grid
> > > >
> > > > ```
> > > > Table 2: what is the meaning of these numbers? e.g., 19(7).
> > > > ```
> > > > This also relates to the two different grids: in parenthesis the result for the small grid, and the other for the large grid (large scale). We will add a word about it.
> > > >
> > > > We hope we have managed to provide answers to all of your concerns and don't hesitate to provide us with some feedback to those.

---

### Official Review · Reviewer_Q14C · 2022-07-26
**Benchmark driven by industry need captures important metrics for method application**

**Rating:** 7
**Confidence:** 3
**Correctness:** Yes, see strengths about the evaluati…

**Strengths:**

* For researchers in physical simulation, it is difficult to find problem settings that are interesting/complex to explore and are guided by industrial need. This benchmark is a very useful and distinct addition to the existing set of canonical PDEs that are often used to benchmark learning methods.
* The evaluation criteria effectively target metrics to compare against existing state of the art methods (traditional simulation), not just among other learning methods. These include how much faster, how well do they generalize and are they physically consistent.
* The results are very well presented (Table 2) and while it is not always possible to threshold performance, I would like to see something like this adopted more widely.
* The datasets and reproducible results are released using a simple API. The authors also set up the benchmark on an automated platform, making comparisons with future work more transparent.


**Weaknesses:**

It would be nice to have a comparison with differently tuned classical simulations. For example, with FEM in the pneumatic tire case, you can vary the mesh coarseness as well as the time step. This would be interesting to study as a trade off between speed and accuracy.


**Additional Feedback:**

N/A

**Clarity:**

The paper is clearly written and the results are presented well.


**Documentation:**

The API for dataset access and reproducing the experiment is well laid out.


**Relation To Prior Work:**

I think there is good discussion of prior work. Similarities exist, but this paper effectively sets itself apart from other learning benchmarks on dynamical systems and other treatments of the subtasks in its stated evaluation criteria and goals.


**Summary And Contributions:**

This paper presents a benchmark for applying machine learning techniques to speed up industrial physical simulation. In addition to laying out two real world problems where speedups are needed, the paper categorizes important metrics for application of solution techniques and presents these results qualitatively. The authors provide an API for access to the data and scripts to reproduce results.

---

> ### Author Response · Authors · 2022-08-08
> **Benchmark driven by industry need captures important metrics for method application**
>
> We would like to thank the reviewer for its thorough review and summarizing the essence of our work and contributions.
>
> This comforts us that our initiative and efforts tries to fill in some well-needed gaps by providing a well-defined benchmark for the ML community geared towards industrial need, on the generic scope of physical simulation usage (not restricted to PDEs-like problems). We appreciate in particular the notice of the visualization and thresholding work as this was a clear methodological achievement for us with the simultaneous objectives of comprehensive and generic benchmarking results.
>
> We eventually hope that with the release of our underlying LIPS framework, new contributions from different communities will easily add new similar tasks from other industrial domains as well as disrupting ML approaches for solving those problems.
>
> Regarding your suggestion of adding a baseline result with a differently tuned physical simulator, we agree this would be a good one and we will add one per domain in our benchmark. For power grids, we will limit for example the number of solver iteration to 1, which should already give rather good accuracy and will be able to see how much speed-up boost we can obtain. For the pneumatic case, we will also do as you suggest.
>
> About the pneumatic datasets, the base data are generated by the notebooks `Benchmark_1_Weight_sustaining_wheel.ipynb` and `Benchmark_2_Rolling_Wheel.ipynb` we provide in the github repository (https://github.com/Mleyliabadi/LIPS/tree/main/getting_started).

---

> > ### Comment · Reviewer_Q14C · 2022-08-24
> > **Response**
> >
> > Thanks for addressing the comments in my review, including the addition of the new baselines. I will change the concerns under documentation and keep the score.

---

### Official Review · Reviewer_Xmsv · 2022-07-27
**Good efforts but does not fit with NeurIPS**

**Rating:** 5
**Confidence:** 2
**Clarity:** The overall writing of the paper seem…

**Strengths:**

Physical simulations are becoming an important interface marrying ML and Science. Fluid dynamics, molecular dynamics, quantum dynamics and neucler power simulators, to name only a few, all fall into the broad concept of "physical simulations". It is desired that more platforms naturally supporting both AI framework and physical simulators could come into use.

**Weaknesses:**

1. The name of "LIPS" seems ambitious, but the current work is limited to pneumatic and power-grid. It is not clear how LIPS can be extended to other physical simulations, and to what kind/domain of physical simulations would be accounted in the future plan for LIPS.
2. The central work is more engineer-related, and there lacks a theoretical breakthrough or any essential connection with ML or AI. So the reviewer would be concerned with its readership in the context of NeurIPS.

**Additional Feedback:**

Currently no more feedback.

**Correctness:**

The reviewer is not an expert in PDE simulations and cannot judge the correctness.

**Documentation:**

Seems fine to the reviewer.

**Ethics:**

No ethical risks were found by the reviewer.

**Relation To Prior Work:**

The authors properly discussed previous work related to some PDE simulations, and some existing industrial applications to power grids.

**Summary And Contributions:**

The authors engineered a platform for users to perform simulations of spcefic kinds, power grid and the pneumatic. ML models are used to fit some controlling parameters used in the simulation engines.

---

> ### Author Response · Authors · 2022-08-08
> **Please reconsider the suitability of this paper w.r.t the "Datasets and benchmarks" track**
>
> We appreciate your time for reviewing our paper and for providing some direct feedback to us that invite us to provide some clarifications.
>
>  Answering your concern if our paper fits with NeurIPS,  we understood that this new track “Datasets and Benchmarks” was there to fill the gap between theory and practice by encouraging groundwork on releasing commonly shared datasets, establishing comprehensive evaluation schemes and providing overall reproducible benchmarks. A paper we referenced ([43]) with  a similar scope was precisely accepted last year for this same track. So, we believe our paper is fully in line with the scope of  this "dataset & benchmark track", even though we recognize that our paper would unfortunately not have fitted historically within the main NeurIPS tracks favoring more ML models or theory.
>
> In fact, the presented work is not only about providing an engineered framework, which is 1 out of 4 contributions, and maybe more a matter of framework development than engineering. Besides that, our main conceptual and methodological contribution was to provide a generic and comprehensive category of evaluation criteria for homogeneous evaluation of ML models across industrial domains concerned with physical simulations. This will drive research towards more impactful and applicable models in the real-world, because today’s research has found limited applicability because of simplistic evaluations, hard to compare and hard to reproduce works.
>
> Note that the tasks in our benchmark suite are not about fitting some parameters within simulation engines, but either emulating or replacing simulation engines: this is a regression task taking the same inputs as physical simulators and inferring all, or part, of the same outputs. As regression might be considered as the 'gold standard' of ML problems, we think that our work is directly connected and targeted to ML and AI. We provide ML baselines to demonstrate that, with some state-of-the art so far in the specific domain literature.
>
>  We also recognize that our work currently only describes results for two use cases. We believed it was more important at this stage to provide this reduced number of use cases to offer an extensive evaluation for each use case rather than less qualitative evaluation over many use cases. Yet, those were chosen to be very different to demonstrate the generic dimension of our evaluation scheme and category of criteria over different physical domains. We hope those categories are comprehensive and generic enough to easily transpose to many other use cases. We are considering for the future additional strong industrial  use cases about aeronautics, energy efficiency in buildings or gas liquefaction.
>
> In the end, our LIPS framework is  not really about supporting the joint development of physical simulations and AI as we understand from your review. This is more the case of NVIDIA Modulus framework and Omniverse overall initiative (which might indeed require a lot of engineering effort and may be difficult to generalize to many domains considering the diversity of simulator implementations to take into account). We can add a word for clarification about this point indeed in the paper. Our framework is rather about offering a generic evaluation scheme with standard benchmark configurations, taking as input datasets and trained models. If our set of criteria is generic and comprehensive enough for all use cases, we think our benchmark suite framework will easily integrate them.
>
> We are keen to take any feedback from you on the genericity and comprehensiveness of our set of criteria, in the scope of industrial physical simulation evaluation, for future improvements of our work. Please let us know also if we were able to answer your concerns or any remaining one you would have.

---

> > ### Comment · Reviewer_Xmsv · 2022-08-23
> > **Appreciation of clarification**
> >
> > The reviewer appreciates the efforts and clarification made by the authors.
> > Since the reviewer is not expert in PDE simulation, this review does not provide valuable reference for technical issues. The reviewer may change the rate if the technical issues raised by other reviewers are well addressed.

---

> > > ### Author Response · Authors · 2022-08-25
> > > **Thank you for your feedback**
> > >
> > > Thank you for your feedback. We have tried to address to our best all other reviewer concerns as well, which answers you can look at for better appreciation.

---

### Comment · Area_Chair_26tB · 2022-08-23
**Please respond to author feedback**

Dear Reviewers,

Please carefully read and respond to all author feedback through the lens of whether it addresses any initial concerns raised.

It is vital for a fair and transparent review process that you read the author’s comments and consider your initial scoring should be adjusted accordingly. If not, please advise the authors why.

Best,

AC

---

### Author Response · Authors · 2022-08-25
**Revised paper uploaded**

Dear reviewers,

A revised version of the paper has been uploaded, that includes most of the feedback you provided.
The main changes are highlighted in blue, we put also a new comparative table (table 2) that better position our work w.r.t. the state of the art. We improved also the references to the supplementary materials for a better understandability of our work, in particular regarding dataset description.

We hope these changes improve our contribution.

---

### Author Response · Authors · 2022-08-25
**Summary of our final revision and contributions**

We would like to thank all reviewers for their precious time in their first review of the paper. We definitely appreciate all comments as it pushes us for clarification and improvements. We now think our original contribution is clarified and better positioned. We did our best effort to answer comments and concerns raised by the reviewers.

Yet, while enthusiastic about an OpenReview process in theory, we found that for the second year in a row for this track it showed some ineffectiveness and we would like to mention two concerns:
- Lack of response and feedback when major concerns are raised. We tried to launch the discussions as early as possible, following the chairs' recommandation, to possibly disambiguate some concerns or find out how to best address them but this rather failed. We nonetheless appreciated the area chair comment.
- Lack of mention of Supplementary Material (or appendix) which might be outlooked in reviews while it contains substantial information and work. While we understand reviewers don’t have time to read it fully, and we try to provide the most necessary information in the main paper, we have hoped it would be a good source of information when looking specifically for some details. We wonder if it is visible enough on OpenReview.

We now summarize our contribution and improvements for the final decision. Revisions in the paper are highlighted in blue to ease any last reading.

As a reminder, we have chosen 2 very different use cases in terms of industrial domain and physics to test the genericity and comprehensiveness of our framework. We have also chosen a use case (power-grid) with already well-established ML research and showed the current limitations to make further meaningful progress without a benchmark. For the pneumatic use case, ML research is very scarce while it offers interesting complexity to be solved with important real-world problems. We hence also give a spotlight to such new problems.

As a follow-up contribution to Reviewer vNMc comment, we have produced a comprehensive comparative related work table (see at the end of the comment and in the revised paper) that best highlights what LIPS really stands for: a unifying and extended evaluation framework contribution. This shows that when ML research is growing over years as in the domain of power grids, it will not have more significant real-world impact if build on heterogenous and incomplete evaluation schemes, making meaningful comparison impossible. It will be interesting to see if we can leapfrog these shortcomings for use cases at an early ML-research stage such as for the pneumatic.

Under Reviewer vgwR comments, we have now explicitly framed 3 corresponding research questions to our contributions and novelty, clarifying the research aspect. The above mentioned table helps answering the first one, while the 2 use cases integration and benchmark result answers the last 2. We have also listed 3 open-research questions to the ML community which should contribute to interesting ML research.

Following-up a suggestion of reviewer Q14C, we are integrating a differently fine-tuned physical simulator baseline, faster but coarser, to challenge how significant ML models speed-up is. We see that in both use cases, the best possible speed-up of such baselines is below a factor 10. There is indeed room for significant ML model contributions.

This also addresses one concern of reviewer vgwR for other baseline. Combined with a reduced-ordered model, a standard Fully-connected NN and a SOTA LeapNet, we think we have covered quite a meaningful set of baselines to highlight the challenge of the power grid benchmarks designed with RTE. We have also made sure we have selected the best challenging reference physical simulator, a concern of reviewer vNMc.
For the pneumatic, as there is no real state-of-the art ML model with a lack of ML research for it so far, the set of baselines is hence a bit reduced, but the benchmark is representative of an industrial challenge to solve, as designed with Michelin. We have tried to provide PINN baseline lately but without much success so far. Yet our role is not to actually solve our benchmark. We believe it is up to the ML research community to better tackle it now.

In terms of benchmark and dataset descriptions, a concern for Reviewer vgwR, we have better referenced pointers to "datasheet for dataset" and appendix sections with already useful information. In particular physical equations and tables about input/output variables, and we added variable dimensions. We have also added some relevant external links for more details about the dataset variables. Every benchmark task now have one corresponding referenced paper in case readers are looking for even deeper understanding.

We hope that overall we were able to provide a meaningful work and benchmark design to the ML community. We will wait for the final decision.
For more details, see also our individual responses to the reviewer.

---

> ### Author Response · Authors · 2022-08-25
> **Related work comparative table that highlights our unifying and extended contributions**
>
> See here the above-mentioned table (colored version in paper) as a clarifying contribution
>
> |                  |                                | evaluation criteria categories |                      |                     |               | impact                  | readability              | environnement setup                                |                                      |                     |                 |
> |------------------|--------------------------------|--------------------------------|----------------------|---------------------|---------------|-------------------------|--------------------------|----------------------------------------------------|--------------------------------------|---------------------|-----------------|
> |                  | **paper reference**                | **ML-related**                   | **industrial readiness** | **ood generalization**  | **PL compliance** | **meaningful thresholding** | **standard comparative viz** | **dataset**                                            | **ref Physical simulator**              | **ML model repository** | **baselines**      |
> | **ML-model papers**  | LeapNet [51]                   | yes                          | no                   | yes                 | no            | no                      | no                       | large, simple prod distributions                   | fast - Hades2 (proprietary)          | maintened           | yes & diverse   |
> |                  | GNS/DSS [13,31,34]             | yes                          | no                   | no                  | partial       | no                      | no                       | data generation shared, simple distributions       | slow - PandaPower (open source)      | unmaintained        | no              |
> |                  | Fast contingency analysis [32] | yes                          | partial              | no                  | no            | no                      | no                       | not shared, realistic prod but simple topo         | slow - PandaPower (open source)      | no access           | yes & diverse   |
> |                  | Physics-informed GNN [35]      | yes                          | no                   | no                  | no            | no                      | no                       | not shared, realistic prod but simple topo         | none                                 | no access           | yes but uniform |
> |                  | Gridwarm [36]                  | yes                          | partial              | no                  | partial       | no                      | no                       | data generation shared, simple distributions       | none                                 | unmaintained        | yes & diverse   |
> | **Benchmark suites** | **LIPS**                           | yes                          | yes                  | yes                 | comprehensive | yes                     | yes                      | large & doc & realistic prod + topo distributions  | fast - LightSimgrid (open source)    | maintained          | yes & diverse   |
> |                  | SimBench [28]                  | no(otimization & heuristics) | yes                  | yes                 | no            | no                      | yes                      | medium & doc & realistic prod but simple topo      | slow - PandaPower (open source)      | no ML               | yes & diverse   |
> |                  | PowerGridLib [29]              | no (optimization)            | partial              | yes                 | partial       | no                      | no                       | small & doc & realistic prod but static topo       | med-speed - PowerModel (open source) | no ML               | yes but uniform |

---

### Meta-Review · Area_Chair_26tB · 2022-09-05

**Recommendation:** Accept
**Confidence:** 4

**Metareview:**

This work introduce a new benchmark to evaluate machine learning techniques for physical simulation. Based on two real-world applications (power grid and pneumatic simulation), the authors clarified the research questions and define evaluation metrics. Existing ML-based methods do not perform well on the proposed datasets. Overall all reviewer agreed that this work proposes a promising benchmark and I think the proposed benchmark can be useful in real-world application. Even though one reviewer has some concerns on details and setups, the authors provided detailed response to address them. For this reason, I'd like to recommend accept.

---

### Decision · Program_Chairs · 2022-09-16

Accept